# SARS-CoV-2 Vaccine Breakthrough Infections of Omicron and Delta Variants in Healthcare Workers

**DOI:** 10.3390/vaccines11050958

**Published:** 2023-05-07

**Authors:** Elisa Regenhardt, Holger Kirsten, Melanie Weiss, Christoph Lübbert, Sebastian N. Stehr, Yvonne Remane, Corinna Pietsch, Mario Hönemann, Amrei von Braun

**Affiliations:** 1Division of Infectious Diseases and Tropical Medicine, Leipzig University Medical Center, 04103 Leipzig, Germany; 2Institute for Medical Informatics, Statistics, and Epidemiology, University of Leipzig, 04107 Leipzig, Germany; 3Central Institution for Occupational Medicine and Occupational Safety, Leipzig University Medical Center, 04103 Leipzig, Germany; 4Interdisciplinary Center for Infectious Diseases (ZINF), Leipzig University Medical Center, 04103 Leipzig, Germany; 5Department of Anaesthesiology and Critical Care Medicine, Leipzig University Medical Center, 04103 Leipzig, Germany; 6Central Pharmacy, Leipzig University Medical Center, 04103 Leipzig, Germany; 7Institute of Medical Microbiology and Virology, University of Leipzig, 04103 Leipzig, Germany

**Keywords:** SARS-CoV-2, vaccine, breakthrough infection, Omicron variant, Delta variant, healthcare workers, transmission

## Abstract

Understanding SARS-CoV-2 breakthrough infections in vaccinated healthcare workers is of key importance in mitigating the effects of the COVID-19 pandemic in healthcare facilities. An observational prospective cohort study was conducted in vaccinated employees with acute SARS-CoV-2 infection between October 2021 and February 2022. Serological and molecular testing was performed to determine SARS-CoV-2 viral load, lineage, antibody levels, and neutralizing antibody titers. A total of 571 (9.7%) employees experienced SARS-CoV-2 breakthrough infections during the enrolment period, of which 81 were included. The majority (*n* = 79, 97.5%) were symptomatic and most (*n* = 75, 92.6%) showed Ct values < 30 in RT-PCR assays. Twenty-four (30%) remained PCR-positive for > 15 days. Neutralizing antibody titers were strongest for the wildtype, intermediate for Delta, and lowest for Omicron variants. Omicron infections occurred at higher anti-RBD-IgG serum levels (*p* = 0.00001) and showed a trend for higher viral loads (*p* = 0.14, median Ct difference 4.3, 95% CI [−2.5–10.5]). For both variants, viral loads were significantly higher in participants with lower anti-RBD-IgG serum levels (*p* = 0.02). In conclusion, while the clinical course of infection with both the Omicron and Delta variants was predominantly mild to moderate in our study population, waning immune response over time and prolonged viral shedding were observed.

## 1. Introduction

On 11 March 2020, the World Health Organization (WHO) declared the coronavirus disease 2019 (COVID-19) outbreak a global pandemic. Just over two years later, on 2 April 2022, 486,761,597 cases had been confirmed around the world, including 6,142,735 deaths [1]. In Germany, there had been 21,357,039 confirmed cases and 129,695 deaths at this point, of which 1,334,260 cases and 14,757 deaths had been reported from the German federal state of Saxony [2]. On 23 March 2022, the level of SARS-CoV-2 infections in Germany reached an unprecedented high, with an incidence of 1934.8 infections per 100,000 people per seven days and 2655.2 infections per 100,000 people per seven days in Saxony [3].

Since the beginning of the COVID-19 pandemic, several vaccines against SARS-CoV-2 have been developed, five of which were approved for use in the European Union (EU) by 2022 [4]. Amongst these, the novel mRNA vaccines proved especially effective in clinical trials [5]. In the EU, 83.1% of the adult population had received the primary course of one of the approved vaccines as of March 2022 [6]. In Germany, at the same time, 85.6% of the adult population had received the primary course of one of the approved vaccines, while compliance in Saxony was considerably lower (approx. 74%) [7]. Healthcare workers in Germany were obliged to provide proof of completed SARS-CoV-2 vaccination or recent convalescence to their employer by March 15 2022 due to an institution-related vaccine mandate based on §20a of the German Protection Against Infection Act (IfSG) [8]. According to the German Hospital Institute (DKI), hospitals reported that 6% of their employees had not provided this proof, implying a vaccination rate of 94% amongst healthcare personnel. At our institution, the Leipzig University Medical Center (Universitätsklinikum Leipzig, UKL), 85.2% of employees had received the primary vaccination course by 14 February 2022 according to internal records.

While the initial results for the vaccine Comirnaty (BNT162b2) by BioNTech/Pfizer showed an efficacy against symptomatic infection of approximately 95% in phase III studies, the appearance of new SARS-CoV-2 variants in combination with waning immunity over time have led to an increased occurrence of SARS-CoV-2 vaccine breakthrough infections [9,10]. A systematic literature review on SARS-CoV-2 mRNA vaccine breakthrough infections in healthcare workers found a very low incidence of 0.011 to 0.001 per 100 individuals at risk [11]. However, the studies included in this review covered a period following the initial vaccine rollout (9 December 2020 to 14 August 2021) and investigated breakthrough infections within the first six months after the primary course of vaccination, and before the emergence of the Omicron variant and its subtypes. At our institution, the number of breakthrough infections in hospital employees increased rapidly with the regional spread of the Delta and Omicron variants, i.e., from October 2021 onwards. Presently, limited data are available on SARS-CoV-2 breakthrough infections during this period. However, answering key questions on SARS-Cov-2 breakthrough infections in hospital employees is of great importance as we expect that the virus will continue to affect our lives considerably for the coming years.

Here, we report the findings of a prospective cohort study at our institution, characterizing clinical, immunological and virological aspects of SARS-CoV-2 breakthrough infections in fully vaccinated medical employees.

## 2. Materials and Methods

### 2.1. Study Design, Setting and Recruitment

An observational prospective cohort study was performed to examine SARS-CoV-2 breakthrough infections in hospital employees fully vaccinated against SARS-CoV-2 at UKL and the Faculty of Medicine of Leipzig University. UKL is a large tertiary care hospital in Saxony, Germany, with 1451 beds across 29 departments and clinics of all specialties. Together, the two institutions have approximately 7000 employees. Recruitment was performed in collaboration with the hospital’s central institution for occupational health. Eligibility criteria included employment at UKL or the Faculty of Medicine, a minimum of two approved SARS-CoV-2 vaccinations received, with the last dose received no less than 14 days ago, and a positive SARS-CoV-2 reverse-transcription PCR (RT-PCR) result diagnosed no longer than 3 days ago. 

### 2.2. Data Collection

Participants were enrolled from 1 October 2021 to 14 February 2022. Data and samples were collected during home visits as all individuals were obliged to self-isolate according to the German Protection Against Infection Act (Infektionsschutzgesetz = IfSG) and the Saxon Corona Protection Ordinance (Sächsische Coronaschutzverordnung = SächsCoronaSchVO). Age, gender, height, weight, occupation, workplace, pre-existing medical conditions, medications, dates and types of SARS-CoV-2 vaccines received, as well as previous SARS-CoV-2 infections, were recorded. Regarding their current infection, the date of diagnosis, type and course of symptoms, probable infection route, number of close contacts (i.e., ≤1.5 m proximity to a SARS-CoV-2-positive individual for ≥ 15 min without the use of personal protective equipment (PPE)), travel history and regional incidence at the time of diagnosis were recorded. An oropharyngeal swab and a serum sample were taken for SARS-CoV-2 RT-PCR, SARS-CoV-2 lineage determination and antibody testing. As previous studies were able to show a high correlation between cycle threshold (Ct) values of RT-PCR results and the actual viral load for Ct values < 30, the term viral load will be used here in reference to measured Ct values for easier reflection of this correlation [12]. All swabs were collected by the same trained investigator to ensure consistency. Following their convalescence, a follow-up interview on the type and course of symptoms was carried out via email. 

### 2.3. Laboratory Methods

#### 2.3.1. SARS-CoV-2 RNA Detection

Oropharyngeal swabs were analyzed with a commercially available SARS-CoV-2 RT-PCR assay for the Alinity M analyzer (Abbott, Chicago, IL, USA) targeting the viral RdRp- and N-genes. 

#### 2.3.2. SARS-CoV-2 Lineage Determination

SARS-CoV-2 lineage was determined through probe-based melting curve analysis targeting known signature mutations in the viral spike protein gene using commercially available primers and probes (TIB MOLBIOL, Berlin, Germany) on a LightCycler 96 instrument (Roche, Basel, Switzerland). Lineage B.1.617 (Delta variant) was determined using assays targeting mutations L452R and P681R. Lineage B.1.1.529 (Omicron variant) was determined using assays targeting mutations S371L/S373P and E484A.

#### 2.3.3. Anti-SARS-CoV-2 Antibody Detection

The concentration of antibodies against the receptor-binding domain (RBD) of the spike protein of SARS-CoV-2 in the sera samples was determined using a commercially available Abbott SARS-CoV-2 IgG II Quant assay on an ARCHITECT i2000SR system (both Abbott, Chicago, IL, USA). To obtain the values for WHO binding antibody units (BAU/mL), the test specific values in arbitrary units (AU/mL) were multiplied by a correction factor of 0.142, following the manufacturer’s instructions. 

#### 2.3.4. SARS-CoV-2 Neutralization Assay

The SARS-CoV-2 neutralization ability of sera samples was determined using clinical isolates of SARS-CoV-2 that correspond to the wildtype strain, lineage B.1.529 and lineage B.1.1.529 (Omicron BA.1). 

Serial dilutions of sera samples were prepared (1:10 to 1:640) using Dulbecco′s modified Eagle′s medium (DMEM, Thermo Fischer Scientific, Waltham, MA, USA) as diluent. Additionally, virus stocks were diluted in DMEM to a final concentration of 2000 TCID50/mL for each of the SARS-CoV-2 strains. Subsequently, 50 µL of each serial serum dilution was mixed with 50 µL virus dilution (corresponding to 100 TCID50/50 µL) in a separate well of a 96-well plate and incubated for 60 min at 37 °C. After the incubation period, 1 × 104 Vero E6 cells suspended in 100 µL of DMEM supplemented with 5% fetal calf serum (FCS, Biochrom, Berlin, Germany) were added to each well. Plates were then incubated for seven days at 37 °C in a humidified incubator at a CO2 concentration of 5%. 

After the incubation period, the supernatant of the cells was removed and 100 µL of neutral red (0.013%) solution was added. Subsequently, the plates were incubated at 37 °C for 60 min. Afterwards, the cells were washed with phosphate-buffered saline (PBS, Thermo Fischer Scientific, Waltham, MA, USA) and fixed with 100 µL formaldehyde (3.7%) (Carl Roth, Karlsruhe, Germany). Formaldehyde fixation was carried out for 30 min at room temperature and the supernatant was discarded afterwards. The integrity of the cell layer was analyzed with an Epoch microplate spectrophotometer (BioTek Instruments, Winooski, VT, USA). All serum samples were tested in duplicate. Neutralizing antibody titers (ND50) were calculated according to the Behrens–Kärber method [13].

### 2.4. Statistical Analysis

For the calculation of differences between the two groups, the Wilcoxon rank sum was used, and the median of the difference between samples of both groups including the confidence interval (CI) based on the Hodges–Lehmann estimator or the continuity-corrected normal approximation was provided, depending on whether ties were absent or present, respectively. One participant with breast cancer was excluded from the statistical analysis, as was one participant with a further infection from laboratory analysis.

To investigate the association between patient characteristics, comorbidities, virological parameters and symptoms, linear regression and Firth’s bias-reduced logistic regression were used for continuous and binary dependent variables, respectively. If not stated otherwise, we used the odds ratio (OR) to quantify the strength of the association. *p*-values were reported as nominal *p*-values considering the level of 0.05 as significant or as multiple-testing corrected *p*-values for the number of tests using the method of Benjamini–Hochberg as indicated [14]. To allow a unified visualization of continuous and binary variables, Spearman’s rho was used in the heatmap plots generated with the R package ‘ComplexHeatmap’ vs. ‘2.10.0’ [15]. The relationship between viral load and anti-RBD-IgG serum levels was investigated in a linear regression model using initial Ct values as the dependent variable and anti-RBD-IgG serum levels as the independent variable following log transformation of both variables. Similarly, log2 transformation was used when investigating the relationship between anti-RBD-IgG serum levels with longer shedding in a Firth’s bias-reduced logistic regression using long vs. short shedding as the dependent variable. Long shedding was defined as the last known positive PCR result ≥ 14 days, and short shedding was defined as the first known negative PCR result ≤ 15 days. Finally, 95% CI for the median were calculated using the R package DescTools 0.99.46. All analyses were carried out in R version 4.1.1 [16].

## 3. Results

### 3.1. Study Population

At the end of the enrolment period, UKL and the associated medical faculty had 6928 employees. Of these, 5906 (85.2%) had received at least two vaccinations against SARS-CoV-2. In total, 571 (9.7%) fully vaccinated employees were diagnosed with a SARS-CoV-2 breakthrough infection during the enrolment period, 81 of whom participated in this study (average age: 34.9 years, median: 33 years, range: 18–62 years). One participant was infected with SARS-CoV-2 prior to enrolment and was, thus, excluded from the analyses. Table 1 shows the participants’ demographics.

### 3.2. Clinical Results

Two participants (2.5%) remained asymptomatic, while most reported between 1 and 12 catalogued symptoms with an average of 6 symptoms reported. As shown in Figure 1, the majority of participants reported rhinitis (*n* = 64, 80%), coughing (*n* = 60, 75%), fatigue (*n* = 58, 72%) and headaches (*n* = 56, 70%), while dyspnea and fever were reported by 29% and 26%, respectively. None of the participants required in-patient care.

Participants with a body mass index (BMI) ≥30 were more likely to experience myalgia/arthralgia (nominal *p* = 0.02) and participants over the age of 50 were more likely to experience coughing (nominal *p* = 0.05), while participants with asthma and/or allergies reported less dyspnea (nominal *p* = 0.04) and sore throat (nominal *p* = 0.04) (Figure 2). Participants with a BMI ≥30 and participants over the age of 50 years also reported a higher average number of symptoms (nominal *p* = 0.36 and nominal *p* = 0.24, respectively). We found a significant association between the Delta variant and the symptom of anosmia (adjusted *p* = 0.000001, OR = 36, 95% CI [8–344]) compared to Omicron breakthrough infections.

### 3.3. Laboratory Results

By day 15 after diagnosis, 34 (42.5%) participants had a negative PCR result and, therefore, were classed as fast-recovering, while 24 (30%) continued to be PCR-positive and were, thus, classed as slow-recovering. A total of 22 (27.5%) were not classifiable due to larger gaps in the testing intervals. In total, 75 (92.6%) participants showed Ct values below 30 during their infection. Six (7.4%) participants continued to show Ct values ≤ 30 after ≥10 days; however, no Ct value ≤30 was detected after ≥14 days. The longest period of viral shedding (PCR positive) observed in our cohort was 41 days and occurred in a participant whose anti-RBD-IgG level was below the detection limit. Though the correlation was not statistically significant, we saw that participants with higher anti-RBD-IgG serum levels tended to shed the virus for a shorter period in both Delta and Omicron breakthrough infections (Delta: *p* = 0.21, OR longer shedding = 0.90, 95% CI [0.66–1.06] when doubling IgG levels, Omicron: *p* = 0.22, OR longer shedding = 0.70, 95% CI [0.31–1.23]). No significant correlation was found between the duration of SARS-CoV-2 PCR positivity and age, variant, neutralizing antibody titers or the time since last vaccination. As shown in Figure 3, Omicron breakthrough infections occurred at higher anti-RBD-IgG serum levels than Delta breakthrough infections (p Wilcoxon = 0.00001, median anti-RBD-IgG Omicron = 1235, 95% CI [771–2404] vs. median anti-RBD-IgG Delta = 138, 95% CI [106–220]).

As shown in Figure 4, the neutralizing antibody titers for the wildtype strain and both variants decreased over time and were strongest for the wildtype strain, intermediate for the Delta variant and lowest for the Omicron variant. Compared with individuals vaccinated ≤100 days ago, the geometric mean of the neutralizing antibody titer was, on average, 73% lower (95% CI [61–82%], *p* < 10^−6^) and 83% lower (95% CI [74–88%], *p* < 10^−6^) in individuals vaccinated 100–200 days and 200–300 days ago, respectively. Furthermore, compared with the neutralizing antibody titer for the wildtype strain, the geometric mean of the titers for Delta and Omicron variants was, on average, 64% lower (95% CI [48–74%], *p* < 10^−6^) and 91% lower (95% CI [87–94%], *p* < 10^−6^), respectively.

When comparing the viral load detected in the oropharyngeal swabs of participants with breakthrough infections within the first 100 days following their last vaccination, Omicron breakthrough infections showed a trend for higher viral loads than Delta breakthrough infections (*p* = 0.14, median Ct difference 4.3, 95% CI [−2.5–10.5]) (Figure 5). Across both variants, the viral load was significantly higher in participants with lower anti-RBD-IgG serum levels (*p* = 0.02). Here, a tenfold increase in the anti-RBD-IgG level resulted in a 5% (95% CI [0.8–19]) increase in the Ct value, on average. 

## 4. Discussion

Hospital employees play a crucial role in upholding medical care and, at the same time, are especially prone to SARS-CoV-2 infection due to the nature of their work. Thus, understanding all aspects of SARS-CoV-2 breakthrough infections in fully vaccinated healthcare personnel is an important factor in the global fight to mitigate the effects of the pandemic, especially in healthcare facilities. Our research examined the clinical course, immune response and virology of SARS-CoV-2 breakthrough infections among fully vaccinated hospital employees. 

We observed an above-average vaccination rate among healthcare personnel at our institution compared to the population of Saxony aged 18–64 years, with vaccination rates of 85.2% and 65.2% (*p* < 0.000001), respectively. Increased awareness of the potentially devastating effects of COVID-19 and easy access in close proximity to the workplace in the hospital’s own vaccination center, along with the anticipation of an upcoming mandatory vaccination for healthcare personnel, likely contributed to the higher compliance.

Nearly 10% of fully vaccinated employees experienced a SARS-CoV-2 breakthrough infection during the enrolment period, highlighting the fact that these infections are a challenge not only for the person infected but also for the employing healthcare facility. Despite our cohort being predominantly young (mean age 34.9 years) and healthy (60.5% reporting no regular medication), most participants reported several symptoms and were, therefore, affected considerably by the disease. A potentially protective effect of pre-existing atopic conditions such as asthma and allergies in COVID-19 patients has been observed in other studies, and eosinophilia and IL-13 activation have been hypothesized as possible mechanisms behind this effect [17]. While not statistically significant in our cohort, previous studies found that obesity and older age put patients at risk of higher COVID-19 morbidity and mortality, and our data support these associations [18]. In comparison, previous surveys found fewer symptoms and a higher ratio of asymptomatic patients, which may have been due to differences in data collection [19]. Data collection through home visits during the first days following diagnosis and with the help of questionnaires may have achieved a more thorough ascertainment at the point of maximum burden of symptoms. Our finding that anosmia was significantly less common in Omicron breakthrough infections compared to the Delta variant is in line with previous studies that compared the clinical presentation of the two variants [20]. While acute and even persistent anosmia seemed to be a hallmark of COVID-19 during the initial phases of the pandemic (corresponding to infections with the wildtype strain, Alpha and Delta variants), the occurrence of this disturbing symptom seems to have reduced as the virus further evolved over time [20].

As shown in Table 1, the source of the infection could not be determined with a high level of certainty in over half of cases in our cohort, which was most likely due to the high SARS-CoV-2 incidence in Saxony during the enrolment period and its effects on the feasibility of contact tracing. Of the 37 participants (45.7%) who were aware of their probable infection route, only 5 (6.2%) stated that they had acquired their infection at the workplace. Our institution had strict hygiene measures in place during the enrolment period, which included the use of full PPE during contact with SARS-CoV-2-infected patients, the continuous wearing of FFP2 masks during face-to-face contact and regular testing of patients and healthcare workers, as well as social distancing during meetings and breaks. Our results indicate that these measures may have been effective in curbing the in-house spread to healthcare workers; however, further investigation will be necessary to determine the best way forward for healthcare facilities, taking into account patient and employer protection, as well as resources and organizational aspects. 

Our study results indicate that even in breakthrough infections, patients may remain infectious for ≥10 days, as indicated by high viral loads in the respiratory samples. Furthermore, higher viral loads correlated significantly with reduced immunological response at time of infection in our study population. Here, further research is needed to conclusively determine the effect of pre-existing immunity through vaccination and/or infection on the viral load and clearance of the virus from the upper respiratory tract, especially for Omicron variants [21]. Interestingly, there were two participants in our cohort with no detectable anti-RBD-IgG serum levels, one of whom shed the virus for 44 days. As this observation is well described for immunocompromised patients, a low immune response and, thus, prolonged viral shedding is less well understood in otherwise healthy individuals [22]. Further studies are needed to determine whether the observation that individuals with lower anti-RBD-IgG serum levels tend to shed the virus for longer can be translated into a targeted booster vaccination strategy especially for healthcare personnel, along with other protective measures. 

In healthcare facilities across Germany, booster vaccinations were rolled out and specifically recommended for healthcare workers during the last three months of the year 2021. Thus, the time since last vaccination was shorter for most of our study participants with Omicron breakthrough infections compared to participants with Delta breakthrough infections, a possible confounding factor in our observation that Omicron breakthrough infections occurred at higher anti-RBD-IgG serum levels compared to Delta breakthrough infections. However, when looking only at individuals with two vaccinations, this trend remains apparent, suggesting that the Omicron variant may be able to achieve symptomatic breakthrough infections at higher anti-RBD-IgG serum levels because of its immunoevasive properties [23]. It has been proposed that antibody serum levels below 2641 BAU/mL significantly increase the risk of developing an Omicron breakthrough infection [19].

The decrease in neutralizing antibody titers over time and the lower neutralization capacity against the Omicron variant observed in our cohort likely played a role in the rapid spread of this variant and the increase in breakthrough infections as Omicron became dominant. The observation that Omicron breakthrough infections tended to coincide with higher viral loads may also underline higher contagiousness and the limited vaccine protection against this variant. In this cohort, both Delta and Omicron breakthrough infections presented with higher viral loads at the time of diagnosis than had previously been reported for breakthrough infections with the Alpha variant [24].

We were able to see a significant negative correlation between anti-RBD-IgG serum levels and the viral load detected in oropharyngeal swabs (*p* = 0.02), suggesting that individuals with higher immune response following vaccination may have been able to control viral replication more effectively. This could have implications for both the severity of disease and infectiousness in vaccinated individuals. Several previous studies have focused on correlations between the severity of disease, viral load and humoral immune response through infection in unvaccinated individuals, one of which found a negative correlation between nasal anti-S antibodies and viral load [25,26,27]. However, the relationship between pre-existing vaccination anti-RBD-IgG serum levels and viral load has not been sufficiently explored. While there was no significant correlation between anti-RBD-IgG serum levels and symptoms in our cohort, this should be re-evaluated in a larger study. 

## 5. Limitations

Our study has the following limitations. Firstly, we were only able to include a subset of all health workers with breakthrough infections (*n* = 81, 14%). We can, however, assume that our cohort reflects the population of healthcare workers at our institution well. Secondly, most Delta breakthrough infections occurred in individuals who had received two vaccines, whereas the Omicron variant was dominant during a time in which most individuals had already received their third (booster) vaccination, as recommended. This has implications, especially on our immunological findings at the time of breakthrough infection. Nevertheless, it also reflects the real-life scenario in which our research took place. The topics of time since booster vaccination, immunological response and circulating variants will continue to be of great interest during the coming years. Lastly, information on reported symptoms included type and course; however, we did not apply a grading system for symptom severity, so correlations between severity and, for instance, viral load or antibody titers were not possible.

## 6. Conclusions

SARS-CoV-2 breakthrough infections pose a challenge for healthcare facilities, including ours, especially during the height of the pandemic. While the clinical course of infection with both the Omicron and Delta variants was predominantly mild to moderate in this young and healthy study population, waning immune response over time and prolonged viral shedding were observed. As SARS-CoV-2 is becoming endemic, high levels of population immunity will hopefully continue to lessen its severity and further reduce the challenges posed for healthcare facilities. In the meantime, further research is needed to determine the best way forward concerning effective protective measures in healthcare facilities during times of high SARS-CoV-2 incidence. 

## Figures and Tables

**Figure 1 vaccines-11-00958-f001:**
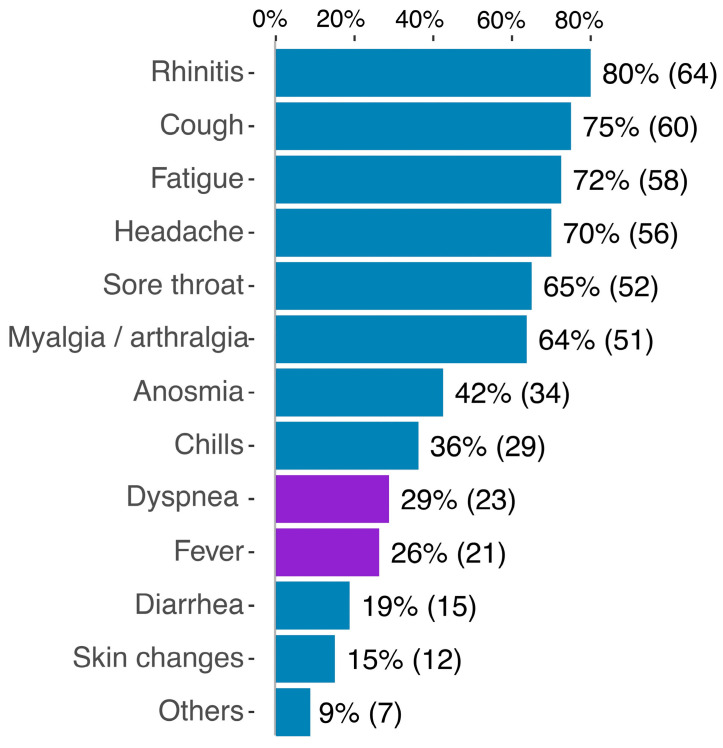
List of reported symptoms in percent of participants (n).

**Figure 2 vaccines-11-00958-f002:**
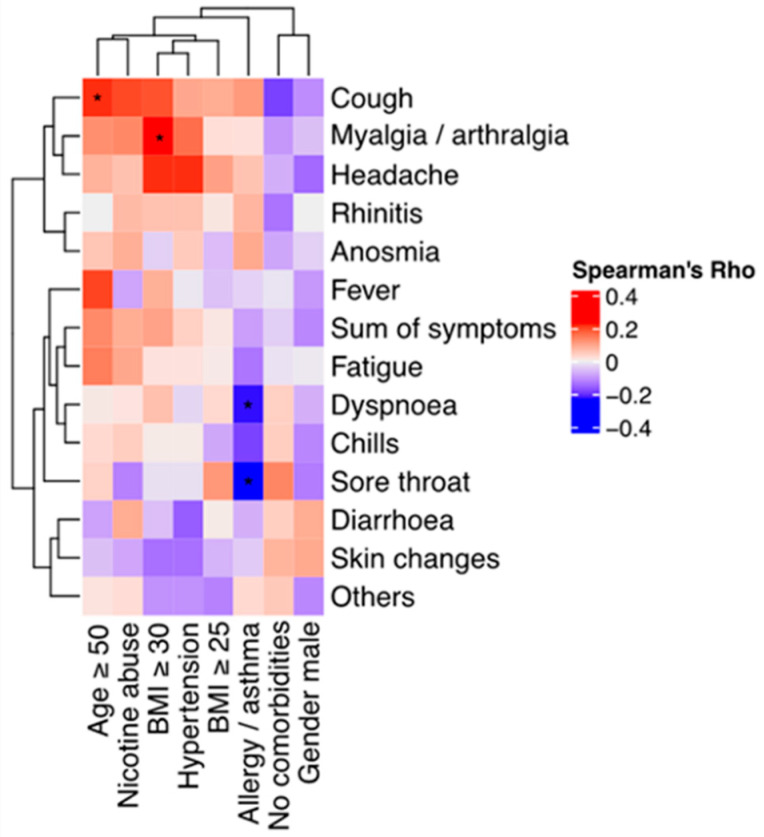
Spearman‘s rank correlation coefficient for participant demographics/comorbidities and reported symptoms (heatmap plots) * *p* < 0.1.

**Figure 3 vaccines-11-00958-f003:**
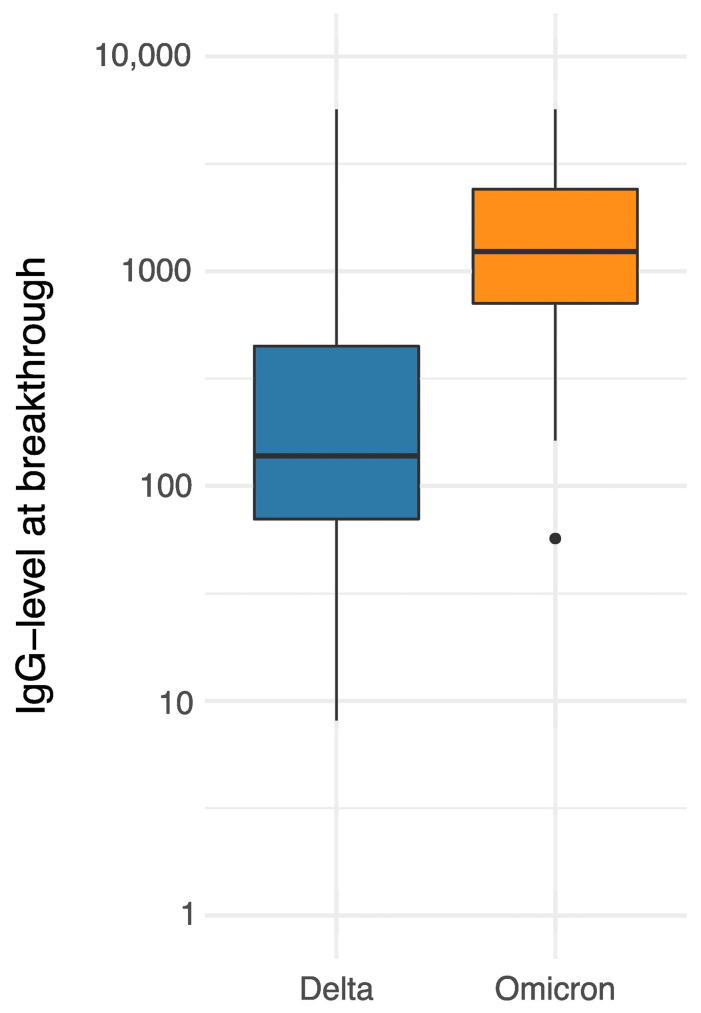
Anti-RBD-IgG serum levels in BAU/mL at breakthrough for Delta and Omicron variants.

**Figure 4 vaccines-11-00958-f004:**
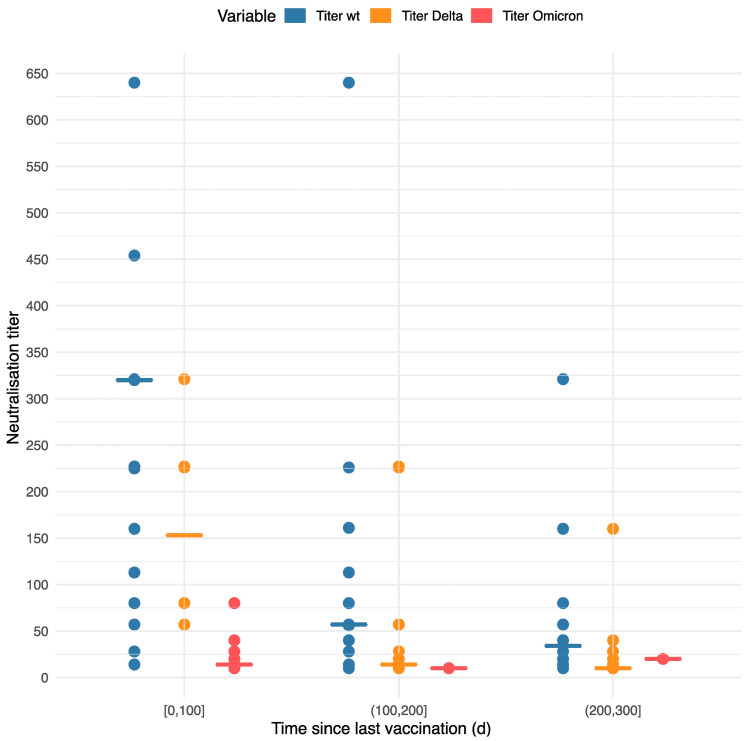
Neutralization titers for wildtype (wt), Delta and Omicron variants, categorized by days (d) since last vaccination (three groups: 0–100 d, 100–200 d, 200–300 d).

**Figure 5 vaccines-11-00958-f005:**
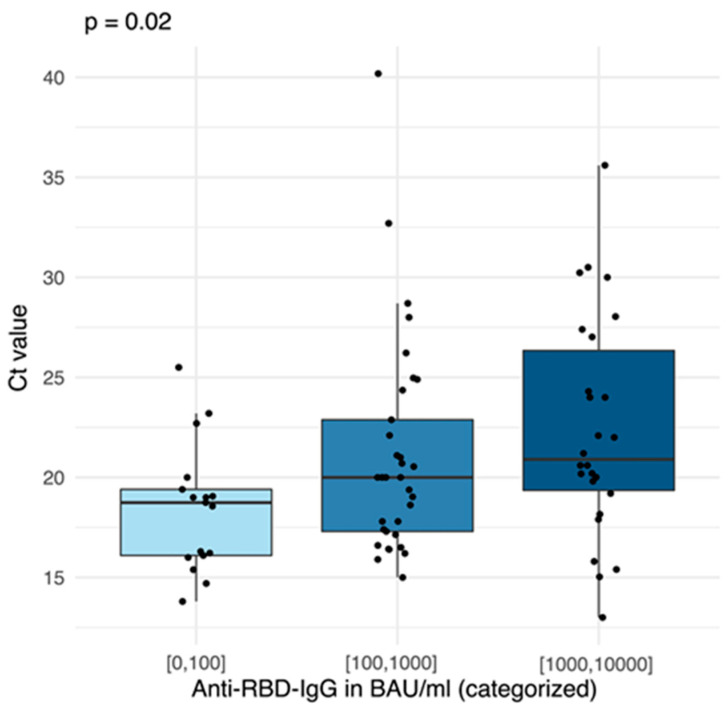
Correlation between Ct value and Anti-RBD-IgG serum level in BAU/mL at breakthrough, categorized in 0–100 BAU/mL, 100–1000 BAU/mL and 1000–10,000 BAU/mL.

**Table 1 vaccines-11-00958-t001:** Study population (*N* = 81).

Demographics	*n* (%)
Sex female/male	56 (69.1)/25 (30.9)
Mean age in years (min, max) ^1^	34.9 (18, 62)
Comorbidities	47 (58.0)
Regular medication	32 (39.5)
Occupation and workplace	
Doctor/nurse/other	17 (21.0)/27 (33.3)/37 (45.7)
ICU/regular ward/other	7 (8.6)/21 (25.9)/53 (65.4)
Patient-facing/non-patient-facing	69 (85.2)/12 (14.8)
Chain of SARS-CoV-2 infection	
Known	37 (45.7)
At workplace/outside of work	5 (6.2)/32 (39.5)
Number and type of vaccinations	
Two/three vaccines	81 (100.0)/33 (40.7)
First vaccine 1/2/3/4 ^2^	62 (76.5)/17 (21.0)/1 (1.2)/1 (1.2)
Second vaccine 1/2/3/4 ^2^	77 (95.1)/3 (3.7)/1 (1.2)/0 (0)
Third vaccine 1/2/3/4 ^2^	32 (39.5)/0 (0)/1 (1.2)/0 (0)

^1^ Numerical variables displayed in this row. ^2^ 1 = Comirnaty, 2 = Vaxzevria, 3 = Spikevax, 4 = Jcovden.

## Data Availability

The data presented in this study are openly available in Zenodo: https://zenodo.org/badge/DOI/10.5281/zenodo.7778728.svg (DOI: 10.5281/zenodo.7778728) (accessed on 27 March 2023).

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
