# Peer review of "SARS-CoV-2 Vaccine Breakthrough Infections of Omicron and Delta Variants in Healthcare Workers"

_vaccines, 2023, doi:10.3390/vaccines11050958_

Round 1
Reviewer 1 Report
Great manuscript - just a few minor recommendations:
Materials and methods:
Page 2 – define UKL
Page 3 – define DMEM and FCS; also add company name, city, country
Page 3 – provide company, city, country for neutral red and formaldehyde
Page 4 – provide reference for R version 4.1.1
Results:
Page 5 – define OR and CI – the odds ratio was not described in the methods section
Page 7 figure 3b – define wt
Reference:
Page 11 – references are double-numbered
Materials and methods:
Page 2 – define UKL
Page 3 – define DMEM and FCS; also add company name, city, country
Page 3 – provide company, city, country for neutral red and formaldehyde
Page 4 – provide reference for R version 4.1.1
Results:
Page 5 – define OR and CI – the odds ratio was not described in the methods section
Page 7 figure 3b – define wt
Reference:
Page 11 – references are double-numbered
Reviewer 2 Report
This article is interesting, but has the following shortcomings;
1. It is obviously inappropriate to represent fig3a and fig3b separately. Figure legends should be merged, with a and b marked on the drawing.
2. All figure legends are too simple for readers to read. It needs to be supplemented by the author.
3. The description of laboratory methods is not clear enough, and the purpose of each method should be described more clearly and in detail.
4. In the result description, a large number of result descriptions are not included in the figure, and these results should be listed in a table for easy reading by the reader. For example, [Compared with individuals vaccinated ≤100 days ago, the geometric mean of the neutralizing antibody titer was on average 73% lower (95% CI [61%-82%], p<10-6), respectively.
When comparing the viral load detected in the oropharyngeal swabs of participants with breakthrough infections within the ffrst 100 days following their last vaccination, Omicron breakthrough infections showed a trend for higher viral loads than Delta breakthrough infections (p=0.14, median Ct-difference 4.3, 95% CI [-2.5-10.5]).
Across both variants, the viral load was signiffcantly higher in participants with lower anti-RBD-IgG serum levels (p=0.02) (Figure 4). Here, a tenfold increase in the antiRBD-IgG level resulted in a 5% (95% CI [0.8-19]) increase of the Ct-value, on average. ], etc.
Reviewer 3 Report
The need for booster vaccinations at a time when covid is now comparable to the common flu is a challenging claim to make, especially in young and healthy persons.
Furthermore, the limitations connected to vaccinations and the variants delta and omicron (as described by the same authors) make assessing the real immunological response of the people investigated unfeasibly.
Additionally, the immune response has been reduced not just in the context of SARS-CoV2, but also in the situation of other viruses (for example, herpes zoster and simplex).
Round 2
Reviewer 2 Report
Authors should emphasize their new findings in the abstract and conclusion of the manuscript, making it easier for readers to read.
Author Response
Dear reviewer, on behalf of all co-authors I would like to thank you again for your comments. Following you suggestion to emphasize our findings in the abstract and conclusion of the manuscript, we have revised the final sentence of our abstract and adapted the conclusions of the main text accordingly. We appreciate your contribution to improving our manuscript and hope our changes meet your expectations.
The following changes have been made:
Abstract, final sentence:
"In conclusion, while the clinical course of infection by both Omicron and Delta variants was predominantly mild to moderate in our study population, waning immune response over time and prolonged viral shedding were observed."
Main text, conclusion:
"SARS-CoV-2 breakthrough infections pose a challenge for healthcare facilities, including ours, especially during the height of the pandemic. While the clinical course of infection by both Omicron and Delta variants was predominantly mild to moderate in this young and healthy study population, waning immune response over time and prolonged viral shedding were observed. As SARS-CoV-2 is becoming endemic, high levels of population immunity will hopefully continue to lessen its severity and further reduce the challenges posed for healthcare facilities. In the meantime, further research is needed to determine the best way forward concerning effective protective measures in healthcare facilities during times of high SARS-CoV-2 incidence."Reviewer 3 Report
A booster vaccine dosage is currently not practical, particularly in healthy patients. There is no longer an emergency situation, and treatment for SARS-CoV2 infection symptoms is standardised. New pharmacological treatments are also available, eliminating the need for a new vaccination campaign.
Author Response
Dear reviewer,
on behalf of the co-authors, I would like to thank you for your additional comment, which has been helpful to us. I believe I missed the point the first time around, please excuse this. Taking your thoughts into account, we revised the last sentence of our abstract focussing this more on our findings directly, as appropriate. Additionally, we revised the conclusions in the main text in the sense of adding the aspect of upcoming endemicity. I truly hope our adaptations are in line with your comment and thank you again. Kindly find the changes in detail below.
Abstract, final sentence:
"In conclusion, while the clinical course of infection by both Omicron and Delta variants was predominantly mild to moderate in our study population, waning immune response over time and prolonged viral shedding were observed."
Main text, conclusion:
"SARS-CoV-2 breakthrough infections pose a challenge for healthcare facilities, including ours, especially during the height of the pandemic. While the clinical course of infection by both Omicron and Delta variants was predominantly mild to moderate in this young and healthy study population, waning immune response over time and prolonged viral shedding were observed. As SARS-CoV-2 is becoming endemic, high levels of population immunity will hopefully continue to lessen its severity and further reduce the challenges posed for healthcare facilities. In the meantime, further research is needed to determine the best way forward concerning effective protective measures in healthcare facilities during times of high SARS-CoV-2 incidence."